# Statin-Induced Myopathy: Translational Studies from Preclinical to Clinical Evidence

**DOI:** 10.3390/ijms22042070

**Published:** 2021-02-19

**Authors:** Giulia Maria Camerino, Nancy Tarantino, Ileana Canfora, Michela De Bellis, Olimpia Musumeci, Sabata Pierno

**Affiliations:** 1Section of Pharmacology, Department of Pharmacy and Drug Sciences, University of Bari “Aldo Moro”, 70125 Bari, Italy; giuliamaria.camerino@uniba.it (G.M.C.); nancy.tarantino@uniba.it (N.T.); ileana.canfora@uniba.it (I.C.); michela.debellis@uniba.it (M.D.B.); 2Unit of Neurology and Neuromuscular Disorders, Department of Clinical and Experimental Medicine, University of Messina, 98122 Messina, Italy; omusumeci@unime.it

**Keywords:** statin, skeletal muscle, ion channels, myopathy, biomarkers

## Abstract

Statins are the most prescribed and effective drugs to treat cardiovascular diseases (CVD). Nevertheless, these drugs can be responsible for skeletal muscle toxicity which leads to reduced compliance. The discontinuation of therapy increases the incidence of CVD. Thus, it is essential to assess the risk. In fact, many studies have been performed at preclinical and clinical level to investigate pathophysiological mechanisms and clinical implications of statin myotoxicity. Consequently, new toxicological aspects and new biomarkers have arisen. Indeed, these drugs may affect gene transcription and ion transport and contribute to muscle function impairment. Identifying a marker of toxicity is important to prevent or to cure statin induced myopathy while assuring the right therapy for hypercholesterolemia and counteracting CVD. In this review we focused on the mechanisms of muscle damage discovered in preclinical and clinical studies and highlighted the pathological situations in which statin therapy should be avoided. In this context, preventive or substitutive therapies should also be evaluated.

## 1. Introduction

The 3-hydroxy-3-methyl-glutaryl coenzyme A (HMG-CoA) reductase inhibitors, also called statins, have a relatively recent history, being discovered in 1970 by Endo [1]. They are effective hypolipidemic drugs that act by inhibiting the production of mevalonate in the biosynthetic pathway of cholesterol. Multicenter clinical trials have demonstrated that these compounds are effective in reducing cardiovascular events and increase the rate of survival in patients [2]. Thus, statins represent the first line therapy in the prevention and treatment of metabolic syndrome [3] and cardiovascular diseases (CVD), that are the primary causes of mortality in the world. Their beneficial role is linked to the reduction of low-density lipoprotein (LDL) cholesterol in blood, resulting from the inhibition of cholesterol synthesis. Thus, based on their capacity to inhibit the HMG-CoA reductase activity and consequently the prenylated protein synthesis, these drugs have been evaluated for their anti-inflammatory and anticancer properties [4,5]. An anti-inflammatory effect of statins has been demonstrated in vitro by the inhibition of cytokines production (i.e., IL-6, TNF-α) [6,7]. For this reason, these drugs have been recently included in treatment protocol for COVID-19 with the aim to control cytokine storm and to control the associated symptoms [8], however, their efficacy is still under study. The clinical use of statin is limited by the adverse effects they can produce. These symptoms are often responsible for the nonadherence or discontinuation of therapy, which may have striking consequences. Indeed, a lower rate of mortality has been observed in patients who were adherent to statins, as compared to the low adherent ones [9]. Different side effects have been reported. The term “statin intolerance” is referred to the incapacity of patients to tolerate a statin dose useful to contrast CVD. Adverse effects included headache, sleep disorders and peripheral neuropathy, gastro-intestinal distress, fatigue, arthritis, increase of liver enzymes, rash, alopecia, erectile dysfunction, gynecomastia [10]. Randomized clinical trials have also shown an increased risk of diabetes [11,12]. However, skeletal muscle symptoms are the most frequent and range from myalgia to life-threatening rhabdomyolysis. Skeletal muscle tissue is disseminated in all the body and its damage may lead to severe consequences [13]. This is what happened with the use of cerivastatin, withdrawn from the market because of an increased risk of rhabdomyolysis, which caused muscle pain and weakness and 52 cases of renal failure and death [14]. Although new drugs are emerging (monoclonal antibodies, antisense oligonucleotides), statins remain the first line option for the treatment of hypercholesterolemia, making it essential to identify the causes of their toxicity. The European Atherosclerosis Society (EAS) Consensus Panel recently described the pathophysiology of statin-associated muscle symptoms (SAMS) and provided a guideline for diagnosis and management of the risk at the aim to ensure the right therapy [9]. In general, SAMS are not common to all statins since different factors may contribute to this condition (genetic, pharmacodynamic, pharmacokinetics, presence of comorbidity, polypharmacy). These recommendations can assist the clinicians in improving adherence to therapy and then the possibility that patients experiencing SAMS may receive the best cholesterol lowering therapy as to minimize CVD risk.

## 2. The Causes of Statin-Induced Myopathy

Although rare, SAMS can be severe. Observational studies describe the occurrence of myalgia in a low percent of patients, with rhabdomyolysis being rarer. Due to SAMS, it is often necessary to reduce the given dose, or to substitute statins, or even to completely stop therapy, thereby precluding the optimal lowering of plasma LDL level [15]. Many studies have reported that the stop of statin therapy leads to an increased occurrence of harmful cardiovascular events [16]. Sometimes, discontinuation of statin therapy may also be due to advice from friends or family members or on information from the lay media. Thus, at the aim to classify these symptoms and to identify the causes, many studies have been undertaken. The clinical description of SAMS covers a wide range of symptoms and the precise classification is still debated. Indeed, an accurate diagnosis of SAMS is hard to assess because symptoms are often subjective and there are no standardized diagnostic tests. The SAMS can arise after few weeks but also after several years of therapy, and the onset of symptoms can occur after an augmentation in statin dose or after the beginning of a concomitant therapy. Interestingly, the SAMS are more frequent in physically active individuals [17,18]. Thus, statin treatment needs to be personalized. Additionally, statins are generally considered as a class, but highlight that differences (pharmacokinetic and pharmacodynamic properties) exist among them and it should be taken into account. A list of adverse effects is available based on the results of clinical trials or on post-marketing surveillance [19]. The classification of adverse effects of statins in skeletal muscle is often different among authors and consensus groups. The definition of SAMS is an unexplained muscle pain, weakness or cramps (myalgia) accompanied or not by a creatine kinase (CK) increase in blood after statin therapy. The most feared statin-induced rhabdomyolysis is a grave form of myopathy with CK typically >40 times the normal value, which generally requires hospitalization, because of massive muscle fiber necrosis. This results in myoglobinuria that can cause a fatal acute renal failure [20].

Different hypotheses have been proposed to understand the causes of myopathy induced by statin. First of all, it has been described that, due to statin effects, the depletion of cholesterol in muscle cell membranes may have an important role in causing myopathy, since its depletion may destabilize sarcolemma and alter ion balance [21,22]. Several lines of evidence have proposed that, due to mevalonate synthesis inhibition, also the reduction of intermediates of the cholesterol biosynthetic pathway, like farnesyl pyrophosphate and ubiquinone (coenzyme Q10) can be harmful for skeletal muscle and mitochondrial function. Indeed, ubiquinone participates in the electron transport chain and in oxidative phosphorylation in mitochondria. It is possible that the decrease of ubiquinone levels causes a loss of ATP and energy production [23]. This is the reason why patients are advised to take CoQ10 while receiving statin therapy. Additionally, the inhibition of prenylation of small GTP-binding proteins like Rho, Ras, and Rac by geranylgeranyl pyrophosphate has a role in the alteration of protein synthesis or organelle biogenesis, signal transduction, and intracellular trafficking [24].

Additionally, the reduction of selenoprotein synthesis, due to the inhibition of the mevalonate pathway, may be partially involved in the failing of myocyte regeneration and in SAMS onset [25]. In addition, statin has been found to inhibit AKT/mTOR signaling pathway, involved in muscle growth during development and regeneration. This may be one of the causes of myotoxicity [26]. The adverse events of statins are dose dependent and different predisposing factors are potentially involved in the generation of myopathy, such as low body mass index (BMI), drug interaction, genetics, female sex, alcohol abuse, hypothyroidism, fibromyalgia, extreme physical exercise, serum Vitamin D deficiency [10]. The adverse events associated with statins occur essentially in patients presenting comorbidities, due to drug interactions [27]. Previous studies found a higher expression of *atrogin-1*, a key gene involved in skeletal muscle atrophy, in humans affected by lovastatin-induced myopathy [28]. The same alteration was observed in vitro in murine skeletal muscle cells exposed to lovastatin. In accord, simvastatin, by impairing PI3K/Akt signaling and FOXO transcription factors, upregulates the genes involved in protein catabolism, carbohydrate oxidation, oxidative stress, and inflammation (*Murf1, cathepsin, myostatin, IL6,* etc.) in treated rats [29]. These animals also show an increase of CK suggesting a direct involvement in the myopathic events.

Statins can also directly target subcellular components and interfere with skeletal muscle function. Indeed, these drugs may alter cytoskeleton integrity with consequent decrease of cell migration, adhesion and viability [30], may disturb endoplasmic reticulum and Golgi vesicular trafficking inducing vacuolation and cell death [31]. Additionally, peroxisome hyperplasia and increased catalase activity has been found due to statin effect, although it is not well known how it can negatively impact on skeletal muscle function [26]. Moreover, in muscle tissue of statin-treated rats an increase of the slow isoforms of myosin heavy chain (MHC) was described, which may suggest a fast-to-slow phenotype shift [32].

A rare form of statin-induced necrotizing autoimmune myopathy (SINAM) has been observed in a low number of patients, in which autoantibodies directed against 3-hydroxy-3-methyl-glutaryl-coenzyme-A reductase (HMGCR) were found [33,34]. In these patients, statin treatment seems to be related to an increase of HMGCR expression, that together with a particular genetic predisposition to autoimmune diseases can induce the production of anti-HMGCR [35]. The histopathological picture showed myofiber necrosis [33,36]. In this condition, the simple discontinuation of statin treatment does not improve clinical symptoms and requires immunosuppressive medication [37,38]. SINAM can occur either after months or years following statin exposure [39,40] and with a prevalence of 1 in 100,000 persons [41]. The observation that many patients with SINAM recover only after immunosuppressive therapy, clearly suggests an immune-mediated pathogenesis [42]. As in other autoimmune diseases, also in the SINAM, the individual genetic susceptibility and the specific environmental triggers are implicated in the pathogenesis. In this case the statins are the “environmental trigger” [43]. It is not excluded that also genetic factors associated to the immune system, as human leukocyte antigen (HLA) are implicated in the SINAM pathogenesis [44].

## 3. Ion Channels as Biomarkers of Statin-Induced Muscle Symptoms of Myopathy

Our previous preclinical studies in the course of many years have shown ion channels as important targets of statin side effects [45,46,47,48,49,50,51]. We have demonstrated that lipophilic statins affect ion channel activity and expression in skeletal muscle. Ion channels are important for muscle function since they control excitability and contractility. A general scheme is reported in Figure 1. In particular, our previous studies show a significant reduction of the resting chloride conductance (gCl) in adult rats treated with high doses of simvastatin, fluvastatin and atorvastatin, likely through a mechanism independent of their capacity to reduce endogenous cholesterol. The resting gCl, sustained by the muscle ClC-1 chloride channel, plays a critical role in the maintenance of resting membrane potential and helps to repolarize the sarcolemma after the action potential, allowing a correct contractile activity and preservation of skeletal muscle integrity. It is known that loss-of-function mutation of the ClC-1 channel leads to Myotonia Congenita, a rare disease with hyperexcitability and impaired relaxation as known signs [52,53,54]. Additionally, skeletal muscle exposure to ClC-1 blockers is known to reduce channel activity producing myotonia in vitro [55]. Thus, it became clear that the reduction of gCl during statin treatment, caused by reduced ClC-1 channel activity, can be detrimental for muscle function. We have also shown that gCl can be further reduced by an increased activity of Protein Kinase C (PKC), that is a potent regulator of ClC-1 able to phosphorylate and close it [47,56,57]. Indeed, the PKC have an important role in the onset of action potential by triggering a reduction of gCl and depolarization [58]. This suggests that the PKC-dependent ClC-1 inhibition at this stage has the physiological role to trigger muscle excitability for movement, but the sustained increased activity of PKC can be dangerous for muscle function. In contrast, resting gCl was not modified by pravastatin treatment, suggesting that the physico-chemical characteristics are important in producing different effects. An increase in total potassium conductance (gK) was also recorded in 30% of the rats administered a high dose of simvastatin (50 mg/kg) and in only 15% of the rats administered a high dose of pravastatin (100 mg/kg). This increase was re-established by in vitro application of glybenclamide, a specific compound able to block the ATP-sensitive potassium channel (KATP), the main potassium channel contributing to the total gK. This evidence suggests that the increase of gK can be related to the reduction of the produced ATP, due to the cholesterol pathway inhibition. Indeed, the KATP channels are involved in cell metabolism and open when ATP level is limited. Indeed, their opening is protective for the muscle. However, recent findings suggest that the opening of KATP, induced by statin may be dangerous in pancreatic beta cells because it may lead to inhibition of insulin release [59].

Lipophilic statins are able to increase the release of calcium from mitochondria and sarcoplasmic reticulum in rat and human skeletal muscle [60,61] which induces PKC-dependent phosphorylation and closure of ClC-1 channel. In support, also other authors have reported that the alterations in Ca^2+^ homeostasis can be responsible for the iatrogenic effects of statins [60,62]. Indeed, acute applications of simvastatin on skeletal muscle fibers from human biopsies triggered an increase of intracellular Ca^2+^ that mostly originates from sarcoplasmic reticulum (SR) [63,64,65,66,67]. Genetic variants within the ryanodine (*RyR*) and dihydropiridine (*DHP*) receptor genes are associated to the vulnerability to statin-associated muscle symptoms [68].

We also found that regulators of gene expression, such as myocyte-enhancer factor-2 (*MEF-2*) and histone deacetylase (*HDAC*) were able to affect ClC-1 and PKC [50,57]. It is hypothesized that PKC-theta, an isoform present in skeletal muscle, co-operates with calcineurin (CN), particularly in slow-twitch muscles, to guarantee MEF-2 transcriptional activation, sustaining the expression of slow muscle typical genes [69] and possibly to maintain a low ClC-1 amount [49]. Thus, the increase of MEF-2 found in statin-treated animals can be the cause of ClC-1 reduction [50,70]. Moreover, it has been shown that PKC activation is also associated with reduced expression of the hepatic organic anion transporter OATP1B1 in membrane and reduced function, increasing blood statin concentration [71].

A proteomic study identified a different pattern of expression in statin treated rats, that suggests an alteration of skeletal muscle function. After statin treatment the oxidative and glycolytic enzymes, creatine kinase and the energy production machinery were downregulated. Additionally, an alteration of proteins responsible for cellular defenses against oxidative stress, such as the heat shock proteins, was demonstrated [48].

Since with advanced age skeletal muscles undergo functional modifications, also in terms of gCl reduction [72], muscle events due to statin therapy can be worsened in this period of life. To validate this hypothesis, we performed a long-term treatment with atorvastatin in 24-months-old rats. Our results show a significant reduction of resting gCl, in parallel with a decrease of ClC-1 channel mRNA and protein expression in aged rats treated with atorvastatin, as compared to treated adult animals. As anticipated above, the increase in MEF-2 in treated animals can be responsible for ClC-1 expression modification. The use of appropriate pharmacological tools (in vitro application of chelerythrine) demonstrate that also the activity of PKC was increased, leading to further decrease of gCl in aged rats treated with the statin. In parallel, a marked reduction of the expression of glycolytic enzymes demonstrates an impairment of muscle metabolism and energy production. These results indicate that a marked reduction of gCl together with an alteration of muscle metabolism coupled to age-related sarcopenia can be responsible for the increased risk of statin-induced myopathy in the elderly.

## 4. Translational Studies: Ion Channel Function and Statin-Induced Myopathy in Patients

Translational studies focus on statin-induced side effects in skeletal muscle of patients in therapy and it is important to understand the causes of the harm. Based on our previous preclinical studies, we evaluated whether ClC-1 channel is modified in muscle biopsies of statin treated patients. We examined patients who experienced myalgia and other typical symptoms, such as hyper-CK-emia after starting statin therapy and we compared the results to those obtained in non-myopathic subjects not using lipid-lowering drugs. Importantly, the analysis of protein expression showed a 40% reduction of ClC-1 protein and an increase of phosphorylated PKC-theta in muscle biopsies of statin-treated patients with respect to untreated subjects, independently from their age and statin type. Real-time PCR analysis showed that despite reduction of the protein, the ClC-1 mRNA was not significantly modified, suggesting posttranscriptional modification (decrease of translation or protein maturation, increased protein degradation). In this regard, Murf-1 is increased suggesting protein degradation; MEF-2 and calcineurin (CN) mRNA were not increased suggesting that in humans they are less involved in ClC-1 expression, indeed ClC-1 mRNA was not modified [51]. GLUT-4 transporter was found to be reduced, suggesting alteration of glucose entry and energy insufficiency. However, the phosphorylated form of AMPK protein was increased, indicating the activation of compensative cytoprotective process [51]. In parallel, the expression of *Notch-1*, a gene implicated in the proliferation of muscle cells, was highly expressed in muscle biopsies of statin-treated patients, indicating effective regeneration. In this regard, it has been found that simvastatin was able to improve arteriogenesis, in case of stroke, particularly by regulating Notch-mediated signaling pathway [73]. Additionally, the expression of *PGC-1-alpha* and *isocitrate dehydrogenase* were increased as well as the activity of citrate-synthase (CS) suggesting the mitochondrial biogenesis [51,74]. Thus, in spite of the evident attempt of the muscle to counteract some of the statin-induced damage, it is not able to counteract the reduction of ClC-1 protein and consequent hyperexcitability of sarcolemma, as well as the energy production deficit that appears to be one of the most important troubles associated with statin-related risk of myopathy in humans. Accordingly, other studies showed reduced respiratory chain activity in mitochondria of patients [64]. A disturbance of the structural integrity of the muscle, with swelling, vacuolization and alteration of the transverse tubules (T-tubules) of sarcolemma was found [75], together with increased expression of the SR ryanodine receptor 3 (RYR3) and Ca^2+^ ATPase 3 (SERCA3) mRNAs [76]. Thus, the identification of a biomarker able to evidence myopathy is crucial to avoid therapy interruption and maybe to counteract statin-induced muscle damage. This study confirms the measure of ClC-1 expression as a reliable clinical test in muscle biopsies (when possible) to detect statin-dependent risk of myopathy (Figure 2).

## 5. Higher Risk with Statin Therapy: Role of Comorbidity, Genetic and Drug Interactions

Numerous risk factors, in terms of age, gender, frailty, genetics, occurrence of other diseases, and polypharmacy may accelerate and/or aggravate myopathy symptoms [9,26,77] (Table 1). Statin-related adverse effects can be more severe in the aged population which uses statins for the prevention of cardiovascular diseases. It has been established that aging process strongly affects skeletal muscle function, causing a loss of muscle mass and functional disability [78,79]. In rodents, senescence of skeletal muscle is associated to a reduction of resting gCl and ClC-1 mRNA expression, an increase of PKC activity along with alteration of calcium homeostasis [72,80,81] and of the glycolytic and oxidative pathways [82]. Thus, all of these alterations, also due to statin action [47], can additionally worsen muscle damage. Moreover, the reduced activity of some metabolizing enzymes, typical of aged subjects, may slow-down drug elimination in these subjects that will have a higher risk to develop side effects.

Numerous studies suggest also a genetic predisposition to statins-intolerance [83,84,85]. Thus, modification of the genes codifying for proteins involved in statin uptake, metabolism, or elimination may confer susceptibility to statin-induced myopathy. In patients with myopathy, high concentrations of statins and their metabolites have been found in plasma, suggesting a pharmacokinetic modification [86]. Indeed, alterations in the metabolic process able to activate the lactone forms of lipophilic statin or to biotransform the acid form by the cytochrome P450 isoforms (CYP3A4, CYP2C9 and CYP2C8) to produce metabolites, may have a role in statin-induced myopathy. In particular, statins are transferred into hepatocytes by the organic anion transporting polypeptide (OATP1B1) encoded by the *SLCO1B1* gene [87]. Some genetic variation of *SLCO1B1* gene are associated to the increased risk of myopathy. Two polymorphisms associated with *OATP1B1* may cause statin pharmacokinetics alteration [88]. Regarding statin metabolism, the CYP3A4 systems are responsible for simvastatin, lovastatin and atorvastatin modification [89], and the inhibition of this system induces an increased risk of muscle complications [90]. The *CYP3A4* show few polymorphisms that can affect drug activity [91]. On the contrary, the polymorphisms of *CYP2D6* are linked to a higher incidence of simvastatin and atorvastatin intolerance [92,93]. Genetic risk factors correlated to the statin-induced myopathy also include modification of genes encoding for plasma membrane *calcium ATPase* [94], or genes involved in creatine phosphorylation [95] or in mitochondrial energy production [96,97,98].

Polypharmacy is another risk factor. Indeed, it is known that concomitant therapies with fenofibrate or ciclosporin can worsen statin-related side effects since they can inhibit the activity of metabolizing enzymes [99,100,101]. Thus, the assumption of the different drugs, especially during aging, should be monitored. However, pravastatin represents an exception because it is mainly biotransformed by a sulfotransferase [102,103].

Based on our previous studies, we deduce the importance to monitor the effects of statin treatment in subjects affected by muscle chloride channel (ClC-1) malfunction. Interestingly, a *CLCN1* gene variant (a heterozygote truncating mutation) encoding for the ClC-1 channel, with loss of function, has been observed to be more frequent in patients with statin-induced myotoxicity than in a healthy population [104], strongly supporting our data. Thus, there are other categories of patients with an increased risk of statin-induced myopathy. For instance, subjects affected by Myotonia Congenita [53], as well as patients with other pathologies in which the alteration of chloride channels induces skeletal muscle malfunction, such as the amyotrophic lateral sclerosis (ALS) [105,106] and Huntington disease [107].

It is known that statin, by reducing cholesterol biosynthesis, may reduce energy stores in the muscle. Actually, patients with an impairment of muscle energy production due to enzymatic malfunction at different levels of either glycogen or lipid metabolisms (i.e., PFK and beta-enolase) [108] may risk a worsening of muscle symptoms during statin therapy. In addition, statin therapy may unmask a metabolic myopathy or may increase susceptibility of asymptomatic carriers to develop symptoms when exposed to statins [109]. Among metabolic myopathies the most frequent types, such as myophosphorylase deficit (McArdle’s disease) and carnitine palmityltransferase 2 (CPT2) deficiency, due to energy failure because of a dysfunction of glycogen and lipid utilization in muscle, add risk for myopathic outcomes [109]. Since the number of patients on statin therapy is expected to grow, a pharmacogenetic investigation may be helpful to evaluate and reduce side effects and to improve the compliance.

Indeed, phosphofructokinase deficiency (Tarui’s disease), a hereditary disorder that alters the ability of muscles to utilize carbohydrates (such as glucose) for energy production, was reported in patients suffering from muscle cramps, exercise intolerance and recurrent episodes of rhabdomyolysis during statin therapy [110]. Similar considerations have to be extended to other metabolic myopathies as muscle β-enolase deficiency, an ultrarare metabolic myopathy due to impairment of terminal glycolysis described in few adult patients with muscle fatigability, myalgia and increased CK values after intense physical exercise [111,112,113]. Elevated level of CK may be also observed in other diseases such as Guillain–Barrè syndrome [114]. Our research pointed out that these metabolic pathways may be affected during statin therapy and are severely altered in aged rats treated with statin [48,50]. The downregulation of mRNA level of beta-enolase, a glycolytic enzyme leading to the formation of phosphoenol-pyruvate (PEP), is a clear indication of the glycolytic suffering, which is particularly evident in the skeletal muscle of aged treated rats, a tissue already compromised by aging process. The downregulation of the glycolytic enzymes determines an energy production failure, and strongly contributes to muscle damage. Interestingly, an increase of Pyruvate kinase M2 (PKM2) isoform was observed in both adult and aged rats treated with atorvastatin. At the moment, it is difficult to understand the reason for PKM2 increase since it has been found in rapidly proliferating tissues and in tumor cells [115]. Its increase can be related to an accumulation of the upstream glycolytic metabolites with a shift of glucose metabolism toward alternative pathways responsible for the biosynthesis of important macromolecules required for increased proliferation. It should be underlined that PKM2 was significantly decreased in untreated aged muscles suggesting a surprising regenerative potential of statins. Accordingly, atorvastatin was shown to improve the post-infarct microenvironment by inhibiting the RhoA/ROCK pathway, and then promoting the survival and therapeutic efficacy of transplanted stem cells [116].

Mitochondrial function was modified particularly in aged rodents treated with atorvastatin. Indeed, a significant decrease in mitochondrial NADH dehydrogenase subunit 5 (mt-ND5) (a part of complex I of mitochondrial respiratory chain) was observed in these animals, again suggesting an impairment of energy production mediated by the oxidative phosphorylation system. The perturbation in energy metabolism and ATP synthesis may have a profound impact on the contractile function during aging. In accord, in treated rats, we observed a slight increase of AMP-activated protein kinase (AMPK) mRNA level, a crucial sensor of energy status, known to be activated during metabolic stress, hypoxia and ATP depletion [50,117]. In support of this idea, the slow-twitch muscles seem to be less affected by statin likely because the elevated number of mitochondria may compensate the energy request.

It is still debated whether the use of statins enhances the risk of new-onset diabetes. Recent large-scale meta-analyses support the concept of a diabetogenic effect of statins, since there are clinical trials reporting changes in glycemia and insulin levels. However, a definitive mechanism has not been elucidated yet [118,119]. Recent studies show a decrease of the expression of GLUT4, able to transport glucose into the cells [51] and reduced insulin secretion with consequent alteration of glucose homeostasis [120]. Previous studies show the opening of KATP channels during administration of high statin doses due to low ATP level, supporting reduced insulin secretion [121]. These effects can aggravate an already compromised situation.

Some evidence suggests that patients undergoing intense physical activity suffer more likely from muscle symptoms with respect to sedentary patients, suggesting that athletes can be more intolerant to lipid-lowering therapy able to amplify the CK increases that commonly occur after vigorous exercise [122,123,124].

However, positive results reported that statins are able to ameliorate muscle dystrophy [13,125]. Simvastatin reduced inflammation in dystrophic (mdx) mice together with a shift toward an oxidative metabolism. Moreover, a reduction of fibrosis was observed, a major cause of functional decline of skeletal muscle in mdx mice [126]. Moreover, some authors claim that statin therapy is associated with a decreased risk of Alzheimer disease, because of their ability to decrease the accumulation of cholesterol and amyloid plaques [120,127], but this aspect needs to be elucidated.

## 6. Management of the Risk of Myopathy with Statin Therapy

Statin can be administered at low doses or once every two days in case of intolerance (diagnosed in 15–30% of treated patients) [10,128]. Alternatively, other drugs can be prescribed, such as ezetimibe, colesevelam, and also nutraceutical compounds to substitute statin therapy [20,129,130,131]. More recently, the proprotein convertase subtilisin-kexin type 9 (PCSK9) inhibitors were introduced in therapy [132,133]. PCSK9 is a protease that degrades hepatic LDL-C receptors. Among these inhibitors, evolocumab and alirocumab are fully human monoclonal antibodies in clinical use. These drugs are injected to inhibit PCSK9, and preserve receptor availability so to reduce circulating LDL-C. The most common adverse effects observed in clinical studies were musculoskeletal manifestations, airway infections, nasopharyngitis, hypersensitivity reactions (pruritus, rash, urticaria) and injection-site reactions. These drugs are considered well tolerated but long-term safety data are still lacking. Since these biotechnological drugs do not interact with the system of the hepatic cytochrome, they may be associated with statins and other drugs. These drugs are recommended in high risk patients, in which LDL-C reduction is not achieved also with the highest tolerated dose of statin and in subjects intolerant to statins treatment [134]. These drugs are also effective in heterozygous FH patients. Another PCSK9 inhibitor is bococizumab, a humanized monoclonal antibody, that was discontinued from use because it induced antibodies production that led to attenuation of the cholesterol-lowering effect [128]. Mipomersen, an antisense oligonucleotide, approved by the FDA in 2013, binds and silences the mRNA encoding for apolipoprotein B (apoB), consequently reducing the apoB-containing lipoproteins. This therapy may induce adverse effects as injection-site reactions, flu-like symptoms, nausea, hepatic steatosis, elevations in serum transaminases, specifically alanine aminotransferase (ALT) and now is not clinically available [135]. Lomitapide is a microsomal triglyceride transfer protein inhibitor, responsible for the assembly and secretion of apo-B-containing lipoproteins in both liver (VLDL) and intestine (chylomicron). Lomitapide treatment results in the impairment of triglycerides secretion from the liver, that can lead to the accumulation of hepatic triglycerides and steatosis. Since lomitapide is an inhibitor of cytochrome P450 3A4, it needs to be monitored because of possible adverse effects when coadministered with statins [136]. Recent drugs include bempedoic acid, inhibitor of the enzyme ATP citrate lyase, involved in the cholesterol synthesis, upstream from the site of action of statins. Unlike statins, bempedoic acid is not active in skeletal muscle, decreasing the possibility of muscle side effects. Volanesorsen a second-generation antisense oligonucleotide administered s.c., is able to reduce apoC-III (a protein present on VLDL, LDL, Lp(a), HDL lipoproteins) levels by specific binding of the corresponding mRNA, and promoting its degradation [135]. Among the drugs still in clinical development there is inclisiran, a small interfering RNA (siRNA) that blocks DNA transcription and the synthesis of the PCSK9 protein in the liver. New monoclonal antibodies include evinacumab, an inhibitor of ANGPTL3, a protein that inactivates a specific lipase, reducing triglycerides and LDL cholesterol, independently from the LDL receptor [137]. Antisense oligonucleotides against Lp(a), a cholesterol-rich lipoprotein, are also under investigation.

Nutraceuticals are food components or derivatives that offer a safe and generally well-tolerated alternative to statin intolerant patients. Red yeast rice (containing monacolin K or lovastatin), berberine, plant sterols, omega-3 polyunsaturated fatty acids, policosanols, polyphenols, flavonoids, can offer some benefit as lipid-lowering agents [137,138].

## 7. Summary and Conclusions

This study provides a summary of the causes responsible for statin-induced myopathy. Ion channels and in particular the ClC-1 chloride channel appears to be a susceptible target for statin action. Since these channels are important for skeletal muscle excitability and contraction their alteration can contribute to myopathy. Considering the wide statins use, it is pivotal to better define the role of statins in exacerbating pre-existing myopathies. An increased risk of myopathy due to statin therapy may occur during aging and/or pathologies characterized by basic alteration of skeletal muscle ion channels. Additionally, oxidative and glycolytic metabolism alteration and sarcopenia make the muscle fibers more sensitive to stress conditions and to myopathy. A scheme of the proposed mechanism is represented in Figure 1. All these findings suggest caution with statin therapy in aged subjects and in predisposed adults (Table 1). The possibility to assess a clear role of these biomarkers in determining skeletal muscle damage by genetic analysis or muscle biopsies may pave the way for preventive therapies. For instance, drugs able to increase chloride channel activity and/or PKC inhibitors, allowing gCl recovery, may reduce the statin-induced muscle damage. In addition, the restoration of the energetic pathways, i.e., through Coenzyme Q10 supplementation or through AMPK-targeted pharmacological strategies, may be useful.

## Figures and Tables

**Figure 1 ijms-22-02070-f001:**
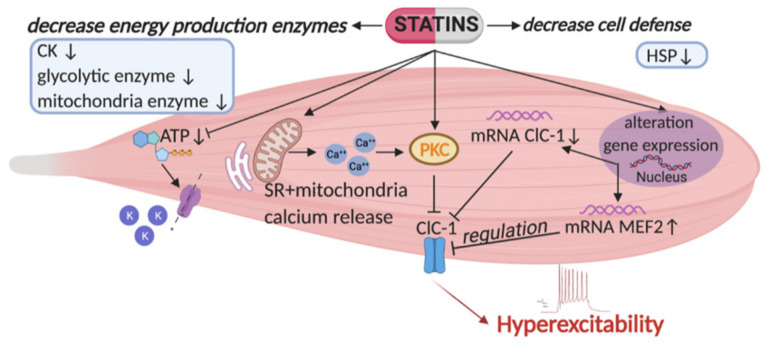
Representation of the major risk factors causing skeletal muscle impairment as found in statin-treated animal model [46,50]. Statins affect the activity and expression of ClC-1 chloride channel, a protein typically present in skeletal muscle and important for its function. Indeed, this channel controls sarcolemma excitability by stabilizing the resting potential, and preventing repetitive runs of discharges after the action potential. As shown, statins reduce ClC-1 gene and protein expression and increase that of the protein kinase C (PKC), which is known to induce phosphorylation and closure of the channel. PKC activation is also sustained by an enhancement of calcium release from the sarcoplasmic reticulum (SR) and mitochondria. This mechanism produces a decrease of resting chloride conductance (gCl), maintained by the ClC-1 channel, leading to a persistent hyperexcitability. It is likely that the modification of ClC-1 can be due to the observed increase of *MEF2* (related to PKC activation and found to be increased during myotonia). Moreover, statin by reducing cholesterol, reduces coenzyme Q (CoQ) and ATP level. This reduction, together with reduction of glycolytic and mitochondrial enzymes, can be detrimental for skeletal muscle function. In addition, ATP reduction leads to the opening of ATP-sensitive potassium channels (KATP) as a compensative mechanism. This is complicated by a decrease of proteins responsible for cellular defenses against oxidative stress, for instance the heat shock proteins (HSP). All these new pharmacological targets may be attractive in the control of statin induced side effects in skeletal muscle. Arrows up and arrows down indicate an increase and a decrease, respectively.

**Figure 2 ijms-22-02070-f002:**
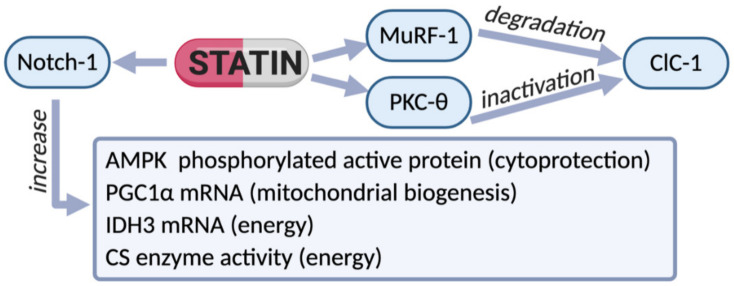
Scheme of the effects of therapy with statins observed in biopsies of patients experiencing myalgia and hyper-CK-emia. In these subjects a reduction of ClC-1 protein was observed, likely due to the increased degradation via MuRF-1 increase. In parallel, an increase of phosphorylated PKC-theta supports ClC-1 inactivation. However, an increased expression of Notch-1 (a protein found to raise during muscle cell proliferation) was observed in statin-treated patients, indicating effective regeneration. In accord, the phosphorylated form of AMPK was increased, suggesting the activation of restorative cytoprotective process. In an attempt to counteract energy deficit, compensative mechanisms were likely activated. Indeed, an increased expression of PGC-1-alpha and isocitrate-dehydrogenase (IDH3), together with increased activity of the enzyme citrate-synthase (CS), suggest mitochondrial biogenesis. Thus, the reduction of ClC-1 protein and consequent hyperexcitability in sarcolemma together with energy deficit seems to be among the most important alterations associated with statin-related risk of myopathy in humans.

**Table 1 ijms-22-02070-t001:** Risk factors that may increase statin-induced muscle symptoms.

Baseline Characteristicsor Genetic Factors	Risk Factors	References
Advanced age	Decreased metabolism (reduced activityof cytochrome P450 isoenzymes);	[89]
Reduction of muscle chloridechannel (ClC-1) activity and expression;	[50,81]
Increased intracellular Ca^2+^;	[80,60]
Energy production defect;	[96,97]
Lower glomerular filtration rate	[16]
Female gender	Different activity of metabolizing enzymes	[26]
Ethnicity andpredisposing genetic variants	Polymorphism and genetic variation of metabolizing enzymes (cytochrome P450) and/or of membranetransporters (i.e., SLCO1B1);	[16][85][26]
Variant in CACNA1H encodingvoltage-dependent calcium channel;	[104]
Variant of ClCN-1 gene encoding for ClC1 channel protein;	[104]
Genetic variants within the ryanodineand dihydropiridine receptor genes	[68]
Vitamin D deficiency	Additional reduction of vitaminD synthesis due to cholesterol reduction	[10]
Low body mass index	Additional cholesterol decrease	[123]
Muscle protein degradation	[98]
Pregnancy	Fetal abnormalities due tocholesterol reduction	[16]
**Exogenous factors**		
Polypharmacy: drugs or food interaction	Drug–drug interaction with inhibitors of cytochrome P450 metabolicisoenzymes (fibrates, immunosuppressantdrugs, cardiovascular antiplatelet oranticoagulant drugs, antimicrobials, and antiviral drugs);	[85][26]

Inhibitors of OATP1B1 (gemfibrozil);	[71,123]
High consumption of grapefruit juice(containing furanocoumarins) thatinhibits CYP3A4 and increases the plasma concentration of statins	
Strenuous exercise	Amplification of the Creatine Kinase (CK) increase that commonly occurs after strenuous exercise	[123,124]
Alcohol abuse	Liver disease (additive transaminasesincrease)	[120,123]
**Energy production defect**		
Mitochondrial gene defect	ATP synthesis reduction	[96,97]
**Comorbidity/Pre-existing diseases**		
Hypertrigliceridemia	Fibrates administration	[101]
Intracerebral hemorrhage	Statin may potentiate the anticoagulant effect of coadministered drugs (i.e., warfarin)	[16]
Amyotrophic lateral sclerosis	Skeletal muscle hypermetabolism;	
Muscle mitochondria energy defect;	
Decrease of nutrients supply to muscle;	[106]
Cholesterol synthesis inhibitionand muscle membrane instability;	
Reduction of muscle chloridechannel (ClC-1) activity andexpression with hyperexcitabilityand alteration of contraction	[16]
[105]
Myotonia congenita	ClC-1 malfunction and skeletal muscle involvement	[53]
Huntington disease	ClC-1 malfunction and skeletal muscle involvement	[107]
Guillain–Barrè syndrome	Additional elevated CK	[114]
Diabetes	GLUT4 mRNA reduction;	
Reduced secretion of insulin	[50]
Inhibition of insulin release due tothe opening of KATP channels in pancreatic beta cells	[120][59]
Hypothyroidism	Hypothyroidism cancause hypercholesterolemia and raised serum CK	[17,101]
Chronic kidney disease	Renal function impairment with increased drug systemic exposure	[16]
Liver disease	Elevated transaminases	[10]
Mitochondrial myopathies	Additive mitochondrial function deficit	[123]
Glycogen storage diseaseGSD V (McArdle disease)GSD VII (Tarui’s disease)GSD XIII	Impairment of glucose andglycogen metabolismPotentiation of myalgia, cramps, fatigueRhabdomyolysis withmyoglobinuria (dark urine)Muscle cramps, exercise intolerance and rhabdomyolysis, symptoms thatcan worsen during statin therapy	[109][110][111][108,112]
Lipid Storage myopathiesCPT2 deficiency	Impairment of lipid metabolismIncreasing of rhadomyolysis episodes	[109]

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
