# Peer review of "Statin-Induced Myopathy: Translational Studies from Preclinical to Clinical Evidence"

_ijms, 2021, doi:10.3390/ijms22042070_

Round 1
Reviewer 1 Report
Dear Authors,
Camerino and Pierno,
Title: Statin-induced myopathy: translational studies from preclinical to clinical evidences
Overall, this review is well written and has exciting figures and one table. In general, the authors covered most of the critical information about Statin's advantages and side effects. However, the author needs to address the following minor points,
Include pathophysiology of statin-associated muscle symptoms (SAMS) in cellular organelles
Mention more details about monoclonal antibodies, antisense oligonucleotides used as a new drug, instead of Statin.
Mention signaling pathways that altered during stain consumption.
What are the effects of Statin on organelle dysfunction in muscle?
Author Response
Statin-induced myopathy: translational studies from preclinical to clinical evidences
Camerino et al.
Response to the Reviewers
Reviewer 1
Overall, this review is well written and has exciting figures and one table. In general, the authors covered most of the critical information about Statin's advantages and side effects. However, the author needs to address the following minor points,
Include pathophysiology of statin-associated muscle symptoms (SAMS) in cellular organelles
Mention more details about monoclonal antibodies, antisense oligonucleotides used as a new drug, instead of Statin.
Mention signaling pathways that altered during stain consumption.
What are the effects of Statin on organelle dysfunction in muscle?
Authors response
First of all we would like to thank the Reviewer 1 for the helpful suggestions and approval.
Below are reported the responses to the Reviewer requests.
To our knowledge little is known about the effect of statin on different cell organelles, apart mitochondria. Indeed, in this review we have described the effects of statin in the modification of electron transport chain and oxidative phosphorylation in mitochondria, which lead to a decrease of ubiquinone and ATP and to adverse muscle symptoms. We also illustrated as sarcoplasmic reticulum is affected by statin action.
However, as suggested by the Reviewer 1, we have included in the new version of the manuscript, some information, already described in the literature, showing the effects of statin in other cellular organelles and found to be responsible for pathophysiology of SAMS (Paragraph 2. The causes of statin induced myopathy).
For instance, simvastatin alters cytoskeleton integrity in cultured cardiac fibroblasts and consequently decreases cell migration, adhesion and viability (Copaja et al., Toxicology, 2012). Moreover, statins disturb Endoplasmic Reticulum to Golgi vescicular trafficking and induce vacuolation and cell death in skeletal myofibers (Sakamoto et al., J Pharmacol Exp Ther, 2011).
Also it has been described that EDL muscles from simvastatin-treated rats exhibited reduced HMGR activity; a 15% shift from the fastest myosin heavy-chain (MHC) isoform IIb to the slower IIa/x suggesting a fast-to-slow phenotype shift (Trapani et al., FASEB J, 2011).
Statins can also directly target other subcellular components and interfere with skeletal muscle function. Peroxisomes are organelles that contain HMG-CoA reductase, the rate-limiting enzyme of cholesterol synthesis (inhibited by statins), and play an important role in isoprenoid biosynthesis. It has been found that lovastatin induces triacylglycerol and phospholipid accumulation in lipid droplets of cultured keratinocytes. These effects are associated with peroxisome hyperplasia and increased catalase activity. Interestingly, peroxisomal hyperplasia was prevented by co-incubation with either low-density lipoproteins or 25-hydroxycholesterol. However, it is not known if statin effects on peroxisomes can negatively impact on skeletal muscle function (Muntean et al., Drug Discov Today, 2017).
2.
As requested by the Reviewer 1, more details are added in this review concerning monoclonal antibodies and antisense oligonucleotides therapy useful to substitute statin in case of adverse effects.
Details about new compounds, side effects and marketing approvation have been included in the revised manuscript. We hope that this satisfy the request of the Reviewer (Paragraph 6. Management of the risk of myopathy with statin therapy).
3.
During statin consumption many signaling pathways are known to be affected. Statin, through the inhibition of mevalonate synthesis, reduces the formation of cholesterol, ubiquinone and dolichols, and impair prenylation of cytoskeleton proteins as well as of small GTPase proteins that have a role in protein synthesis or organelle biogenesis, signal transduction and intra-cellular trafficking (Bonetti et al., Eur Heart J, 2003).
In addition, statin has been found to inhibit AKT/mTOR signaling pathway, involved in muscle growth during development and regeneration. This may cause myotoxicity (Muntean et al., 2017). As suggested by the Reviewer we have added these important informations in the new version of the manuscript (Paragraph 2. The causes of statin induced myopathy).
4.
Thanks to the Reviewer 1 we have added information on the role of statin in producing the myotoxicity through their action on most of the muscle cell organelles (mitochondria, cytoskeleton, endoplasmic reticulum, etc.) (Paragraph 2. The causes of statin induced myopathy).
Reviewer 2 Report
The review is up-to-date and highlights interesting findings related to statin-induced muscle symptoms. Although I apreciated the manuscript, the plethora of grammar mistakes has the tendecy to distract the attention from the scientific information. Thus, extensive English proofing needs to be done.
Regarding the scientific information included, there are a few remarks:
line 28 - I would suggest to say that statins have a "relatively" recent history, as they have been discoveread over 50 years ago by "Endo"
line 68 - there is a repetition of SAMS, if it was abreviated once, then the abreviation should be used further
line 86 - replace "post-marketing experience" with "post-marketing surveillance"
lines 122-124 - reformulate phrase as is quite ambiguous
line 175 - figure 1 legend needs a reference for the "statin-treated animal model" that is mentioned at the begining
line 298 - "SCLO1B1" should be replaced with "SLCO1B1"
Table 1 - "Guillain-Barre syndrome" not "Guillain barre"
Concerning statin-immune-mediated necrotizing myopathy, an excellent recent review could also be included, DOI: 10.1080/1744666X.2018.1440206
Author Response
Statin-induced myopathy: translational studies from preclinical to clinical evidences
Camerino et al.
Response to the Reviewers
Reviewer 2
The review is up-to-date and highlights interesting findings related to statin-induced muscle symptoms. Although I apreciated the manuscript, the plethora of grammar mistakes has the tendecy to distract the attention from the scientific information. Thus, extensive English proofing needs to be done.
Regarding the scientific information included, there are a few remarks:
line 28 - I would suggest to say that statins have a "relatively" recent history, as they have been discoveread over 50 years ago by "Endo"
line 68 - there is a repetition of SAMS, if it was abreviated once, then the abreviation should be used further
line 86 - replace "post-marketing experience" with "post-marketing surveillance"
lines 122-124 - reformulate phrase as is quite ambiguous
line 175 - figure 1 legend needs a reference for the "statin-treated animal model" that is mentioned at the begining
line 298 - "SCLO1B1" should be replaced with "SLCO1B1"
Table 1 - "Guillain-Barre syndrome" not "Guillain barre"
Concerning statin-immune-mediated necrotizing myopathy, an excellent recent review could also be included, DOI: 10.1080/1744666X.2018.1440206
Authors response
We would like to thank very much the Reviewer 2 for precious advices that allowed us to improve our manuscript.
As requested we have ameliorated English language by asking to a mother language lecturer.
Also we have made all the correction requested:
Line 28 “relatively” was added. We agree that statins have a "relatively" recent history.
Line 68. The abbreviation was placed where needed
Line 86 "post-marketing surveillance" was placed
Line 122 we have reformulated the unclear phrase, and we hope that now it is better:
A rare form of statin-induced necrotizing autoimmune myopathy (SINAM) has been observed in a few number of patients, in which were found autoantibodies directed against 3-hydroxy-3-methyl-glutaryl-coenzyme-A reductase (HMGCR) [33,34].
Figure 1. the references 46,50 have been added for the "statin-treated animal model"
Line 289 "SLCO1B1" corrected
Table 1 "Guillain-Barre syndrome" corrected
We thank the Reviewer 2 for the suggested reference (34), added to the new version of the manuscript, that have enriched the bibliography of our review.
Round 2
Reviewer 2 Report
Minor coments:
line 11- Statins are the most....(plural must be used)
line 13- leads (muscle toxicity leads to....)
line 22 - should also be evaluate
line 33 - statins represent
lines 39-40 - these drugs have been evaluated for their anti-inflammatory and anti-cancer properties
line 41 - please detail which cytokines have been demonstrated to be inhibited by simvastatin (ref 6 and 7)
line 44 - statins
line 45 - responsible for the...
lines 46-47 - delete "on benefits"
line 52 - liver enzymes
line 67 - delete "factors"
line 76 - statins
line 82 - covers
line 89 - needs to be...
line 95 - muscles
line 100 - replace "because the massive muscle.." with "because of massive muscle.."
line 103 - effects
line 113 - replaced "maintained" with "receiving"
line 116 - "has a role" not "have a role"
line 121 - statins
line 124 - the adverse events associated with statins
line 126 - drug interactions
line 130 - a higher
line 136 - these animals also show
lines 144-145 - replace "Moreover, it has been described that in muscles of statin-treated rats an increase of 144 the slow isoforms of myosin heavy chain (MHC) suggest a fast-to-slow phenotype shift" with "Moreover, in muscle tissue of statin-treated rats an increase of the slow isoforms of myosin heavy chain (MHC) was describes, which may suggest a fast-to-slow phenotype shift"
line 148 - replace "few number of patients" with "low number of patients"
line 148 - move "were found" at the end of the phrase
line 156 - replace "the simply discontinuation of statin treatment" with "the simple discontinuation of statin treatment"
line 162 - replace "As in the others autoimmune diseases" with "As in other autoimmune diseases"
line 181 - replace "lead to Myotonia Congenita" with "leads to Myotonia Congenita"
line 186 - we have also shown
lines 196-197 delete "with"
Figure 1 - statins - decrease energy production enzymes
line 209 - replace "Statin affects" with ""statins affect"
line 216 - produces
line 222 - leads
line 225 - targets
line 305 - appears to be
line 309 - therapy with statins
line 334 - replace "assume" with "use"
line 369 - represents
line 396 - alters
line 397 - suffering from muscle...
line 404 - replace "involved" with "affected" or "altered"
line 433 - please revise the expression "oxidative muscles"
line 456 - "amyloid" without "s"
line 457 - replace "this need to be better elucidated" with "this aspect needs to be elucidated"
line 478 - statins
line 479 - Bococizumab was not obtained so recently, in fact it was withdrawn from the market in 2016, so I would suggest deleting "recently obtained ..." and reformulating the phrase
line 482 - mipomersen is not so recent, it was developed in 2008 and approved by the FDA in 2013. Liver
line 493 - it needs to be
line 504 - inactivates
line 514 - replace "appears a susceptible 514 parameter to statin action." with "appears to be a susceptible tagert for statin action."
line 517 - statins
Author Response
Response to Reviewer 2 Comments
Minor coments:
line 11- Statins are the most....(plural must be used)
line 13- leads (muscle toxicity leads to....)
line 22 - should also be evaluate
line 33 - statins represent
lines 39-40 - these drugs have been evaluated for their anti-inflammatory and anti-cancer properties
line 41 - please detail which cytokines have been demonstrated to be inhibited by simvastatin (ref 6 and 7)
line 44 - statins
line 45 - responsible for the...
lines 46-47 - delete "on benefits"
line 52 - liver enzymes
line 67 - delete "factors"
line 76 - statins
line 82 - covers
line 89 - needs to be...
line 95 - muscles
line 100 - replace "because the massive muscle.." with "because of massive muscle.."
line 103 - effects
line 113 - replaced "maintained" with "receiving"
line 116 - "has a role" not "have a role"
line 121 - statins
line 124 - the adverse events associated with statins
line 126 - drug interactions
line 130 - a higher
line 136 - these animals also show
lines 144-145 - replace "Moreover, it has been described that in muscles of statin-treated rats an increase of 144 the slow isoforms of myosin heavy chain (MHC) suggest a fast-to-slow phenotype shift" with "Moreover, in muscle tissue of statin-treated rats an increase of the slow isoforms of myosin heavy chain (MHC) was describes, which may suggest a fast-to-slow phenotype shift"
line 148 - replace "few number of patients" with "low number of patients"
line 148 - move "were found" at the end of the phrase
line 156 - replace "the simply discontinuation of statin treatment" with "the simple discontinuation of statin treatment"
line 162 - replace "As in the others autoimmune diseases" with "As in other autoimmune diseases"
line 181 - replace "lead to Myotonia Congenita" with "leads to Myotonia Congenita"
line 186 - we have also shown
lines 196-197 delete "with"
Figure 1 - statins - decrease energy production enzymes
line 209 - replace "Statin affects" with ""statins affect"
line 216 - produces
line 222 - leads
line 225 - targets
line 305 - appears to be
line 309 - therapy with statins
line 334 - replace "assume" with "use"
line 369 - represents
line 396 - alters
line 397 - suffering from muscle...
line 404 - replace "involved" with "affected" or "altered"
line 433 - please revise the expression "oxidative muscles"
line 456 - "amyloid" without "s"
line 457 - replace "this need to be better elucidated" with "this aspect needs to be elucidated"
line 478 - statins
line 479 - Bococizumab was not obtained so recently, in fact it was withdrawn from the market in 2016, so I would suggest deleting "recently obtained ..." and reformulating the phrase
line 482 - mipomersen is not so recent, it was developed in 2008 and approved by the FDA in 2013. Liver
line 493 - it needs to be
line 504 - inactivates
line 514 - replace "appears a susceptible 514 parameter to statin action." with "appears to be a susceptible tagert for statin action."
line 517 - statins
Authors Response
We would like to thank very much the Reviewer for the precious help and to have ameliorated our manuscript. We agree with all the comments and as he/she suggests we have done all the corrections needed.
Line 11-33. The plural and “s” were corrected
Line 39-40. The phrase has been corrected
Line 41. In the new version of the manuscript we have added which kind of cytokines are involved (IL-6, TNFα)
Line 44-136. The corrections have been made
Lines 144-145. The phrase has been replaced
Line 148-197. We agree and corrected as suggested
We have corrected the Figure 1 as suggested
Line 209-404. The corrections have been made
Line 433 “oxidative muscles” has been replaced with “slow-twitch muscles”
Line 456-478. We have followed the suggestions of the Reviewer and modified the text accordingly
Line 479 and 482. We agree with the Reviewer that bococizumab and mipomersen are not recent drugs. We have corrected the text and reformulated the phrase as suggested
Line 493 - 517. We have made the corrections suggested
We would like to thank again the Reviewer and we hope that now the manuscript will fulfill the requested revision.